# Designing optimal tests for slow converging Markov chains

**Cliff Stein** [1]    **Pratik Worah** [2]

## Abstract

We design a Neyman-Pearson test for differentiating between two Markov Chains using a relatively small number of samples compared to the state space size or the mixing time. We assume the transition matrices corresponding to the null and alternative hypothesis are known but the initial distribution is not known. We bound the error using ideas from large deviation theory but in a *non-asymptotic* setting. As an application, using scRNA-seq data, we design a Neyman-Pearson test for inferring whether a given distribution of RNA expressions from a murine pancreatic tissue sample corresponds to a given transition matrix or not, using only a small number of cell samples.

## 1. Introduction

Markov chains underlie many natural phenomena ranging from gene expression mechanisms to stochastic gradient descent algorithms. Given just the empirical distribution from a sample of observations and the transition matrices of two finite state Markov chains, it is natural question to ask whether one can identify which Markov chain resulted in the sample? Such questions are important because they underlie genomic tests, but can be difficult to answer as the initial distribution is unknown, the Markov chains mix slowly, and the number of samples used in the empirical distribution is small. In this note, we present such a hypothesis testing algorithm using a modified empirical log-likelihood, and explain the underlying intuition behind the modification.

A Neyman-Pearson hypothesis test consists of comparing the empirical log-likelihood (equivalently the hypothesis test score) with a fixed constant, and accepting or rejecting the null hypothesis based on the outcome (see chapter 3 in (Dembo & Zeitouni, 1998)). We design a modified Neyman-Pearson test that works with a small number of

---
[1]Google Research and Columbia University, USA [2]Google Research, USA. Correspondence to: Pratik Worah <pworah@google.com>.

*Workshop on Interpretable ML in Healthcare at International Conference on Machine Learning (ICML)*, Honolulu, Hawaii, USA. 2023. Copyright 2023 by the author(s).

samples. We provide an overview of our theoretical results in Section 2. In particular, Equation 2 contains the score calculation associated with our modified Neyman-Pearson test, which is the heart of our hypothesis testing algorithm. The basic idea is to project the empirical distribution to a lower dimension. Although the projection may increase the total error by disregarding some information from the sample, it also increases the reliability of the hypothesis test score by filtering out initial state effects.

*Application to single cell data*: As an application, we use the scRNA expression data from murine pancreatic cells (Bastidas-Ponce et al., 2019) to compare our projected hypothesis testing algorithm with the usual hypothesis test (which requires a much larger number of samples). We use the empirical distribution of Cpe expressions calculated from a sample of 10-20 beta cells, and ask if we can reliably say whether the sample came from (1) beta cells vs alpha cells and (2) beta cells vs ductal cells. Since Cpe is involved in hormone secretion, we expect (1) to be much harder to test than (2). Indeed that is the case: Our projection based test can not succeed at (1) but does seem to succeed at (2). Figures 1 and 2 summarize our initial results and Section 3 contains further details about the experimental application.

*Related Work*: Neyman-Pearson tests are well-known in statistics (Dembo & Zeitouni, 1998). The assumption of large sample size is their major drawback for slow mixing chains. The paper (Sun et al., 2006) investigates the closely related problem of computing the fastest mixing Markov chain supported on a graph. In principle, that may also help reduce the sample size required but their techniques are very different from ours.

## 2. Overview of our theoretical results

Suppose $S_n := s_1 s_2 ... s_n$ is a realization of an ergodic irreducible Markov Chain $M$ on a finite discrete alphabet $A := \{1, 2, ..., m-1, m\}$ with time-steps in $[n]$, where $n = o(m)$. Let $P$ denote its transition matrix. We assume that $P$ is known, and while we assume that the sequence $S_n$ itself can not be observed, the empirical distribution $\hat{\mu}_{S_n} := \frac{1}{n} \sum_{\substack{j \in [n], \\ i \in [m]}} \delta_i(s_j)$ of $M$ can be observed. We also assume that the initial distribution $p$ of $M$ is unknown. Given $\hat{\mu}_{S_n}$, and two candidate transition matrices $P_1$ and $P_2$, our goal is to design a Neyman-Pearson test that either rejects the null

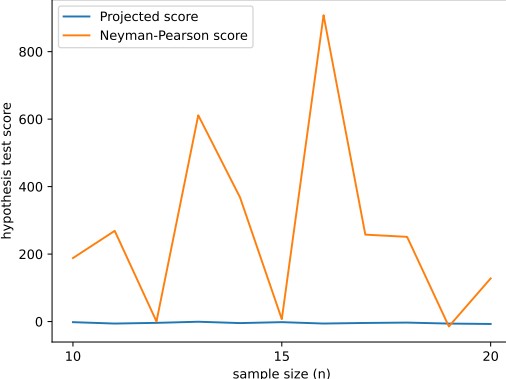
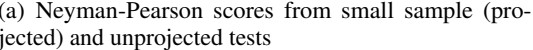
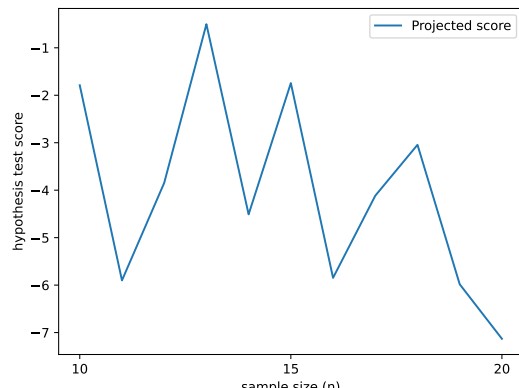

(a) Neyman-Pearson scores from small sample (projected) and unprojected tests

(b) Zoom-in of Neyman-Pearson score from projected (small sample) tests only

*Figure 1.* High positive values of the score suggest that the test empirical distribution is from beta cells (the ground truth) while high negative values suggest the distribution came from alpha cells. Note that the unprojected or unmodified Neyman-Pearson scores are high positive but very variable (as can be expected for the small sample size) but our projected scores are much more stable – note the $y$-range on the right figure above, also compare Figure 2.

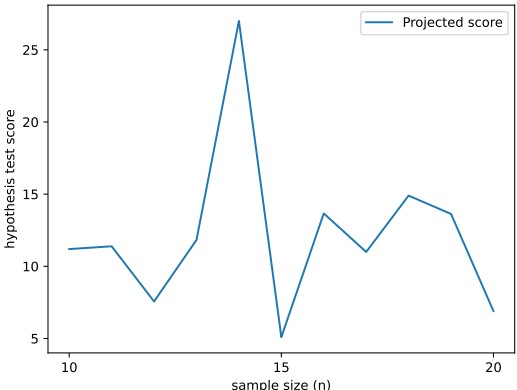

*Figure 2.* High positive values of the score suggest that the empirical distribution is from beta cells (the ground truth). Here we can distinguish between the two cell types (beta cells vs ductal cells) using just 10-20 cell samples of Cpe expressions.

hypothesis $H_0 : \hat{\mu}_{S_n} \sim P_1$ or the alternative hypothesis $H_1 : \hat{\mu}_{S_n} \sim P_2$, where $\sim$ denotes that the measure came from the given Markov chain.

Let $\tau$ be the $\varepsilon$ mixing time of Markov chain $M$. Suppose $n \ll \tau, m$, i.e., we are not assured that $M$ mixes in the time we observe it. In that case, it is natural to ask, what can we say about the properties of the empirical distribution $\hat{\mu}_{S_n}$ of $M$? For example, given a subset of measures on $A$, say $\Gamma$, can we estimate $\mathbb{P}(\hat{\mu}_{S_n} \in \Gamma)$ with a reasonable accuracy?

One way to estimate the probability is to design a coarser alphabet, obtained by grouping subsets of $A$, such that the empirical distribution on the coarser alphabet converges

much more rapidly. An immediate brute force heuristic algorithm to find such a coarser alphabet is: Let $\pi$ be the principle eigenvector of $P$, and $\alpha > 0$ be such that $\forall x, y \in [m], \; P(x, y) \geq \alpha \pi_Z(y)$. Then the $\varepsilon$ mixing time is at most $\frac{\log(1/\varepsilon)}{\log(1/(1-\alpha))}$. Therefore, we can simply group alphabets such that the resulting "Markov chain" corresponding to the coarser alphabet maximizes the $\alpha$ above, then we may obtain a stable empirical distribution in a much shorter time.

There are at least two issues with the above brute force approach:

1. Non-linearity: it is unclear if one can always represent the stochastic process on the coarser alphabet as a Markov Chain (in general the answer is no).

2. Efficiency: there are exponentially many groupings of alphabets to try.

Fortunately, it turns out that we don't need to try all groupings of $A$ if we are satisfied with a near optimal solution. We will define our coarse alphabet via a linear projection, which will allow us to solve the problem via alternating minimization type algorithms.

Given a probability vector $q$ supported on $A$ ($|A| = m$), and a $m \times d$, where $d \leq m$, stochastic matrix $Z$, define the *projection* $\text{proj}_Z(q) : \mathbb{R}^m \to \mathbb{R}^d$ as $\text{proj}_Z(q) := Z^T q$.

In particular, if $d = m$ and $Z = \text{Id}$, then $\text{proj}_Z$ is just the PDF of the original distribution $q$. However, for $d < m$, if the columns of $Z$ are set appropriately (so that each column sum is the same or roughly the same), then we get a *coarse PDF* supported on a smaller size alphabet, only

approximating $q$. We have clearly lost information about our distribution, and that can lead to a higher error in any Neyman-Pearson test. On the other hand, the resulting Neyman-Pearson test using the coarse PDF may be more robust to variation in the initial state. Below we sketch how to quantify the gains from each side, using large deviation techniques.

Recall that, the Gärtner-Ellis theorem (see (Dembo & Zeitouni, 1998)) allows us to asymptotically estimate the large deviation probability $\mathbb{P}(\hat{\mu}_{S_n} \in \Gamma)$, for the empirical distribution of a given Markov chain. That's our starting point, since our projections are linear transformations of the empirical distribution. The main issue is the asymptotic (in $n$) nature of the bound in the Gärtner-Ellis theorem. Typically, $n$ is assumed to be very large – much larger than the mixing time. Throughout, we will assume $n$ is only *moderately large*, i.e., $n \to \infty$ but $n = o(m)$, where recall that, $m$ is the size of the our finite Markov chain state space.

In a full version, we show a non-asymptotic version of Gärtner-Ellis theorem, i.e., its bounds hold for $n = o(m)$. For this overview, we will concentrate on only the upper-bound. In a full version, we show that $\frac{1}{n} \log \mathbb{P}(Z^T \hat{\mu}_{S_n} \in Z^T \Gamma)$ can be upper bounded by a sum of three terms:

1. The log metric entropy term equals: $\frac{1}{n} \log N(Z^T \Gamma, r)$; where $N(E, r)$ denotes the *metric entropy* of a subset $E$ of a metric space, i.e., the number of balls of a radius $r$ required to cover $E$. Note that $Z^T \Gamma$ is the action of $Z$ on a subset of the space of measures and thus $N(Z^T \Gamma, r)$ can be exponentially large in $d$.

2. The rate function term which equals: $\sup_{\lambda \in \mathbb{R}^d} \left( \langle \lambda, Z^T q \rangle - \log \rho_Z(\lambda) \right)$; where $\rho(P_{\lambda, Z})$ is the principal eigenvalue of $P_{\lambda, Z}$ for the matrix $P_{\lambda, Z}(i, j) := P(i, j) e^{\langle \lambda, Z^T 1_j \rangle}$. [1] This is the usual rate function term found in large deviations results, and would be the only surviving term if we were to assume $n \gg \tau$ (the mixing time).

3. The initial state term which can be upper bounded by: $\frac{1}{n} \cdot \max_{i \in [m]} \{ \log(\frac{p(i)}{\pi_Z(i)}), \log(\frac{\pi_Z(i)}{p(i)}) \}$; where $\pi_Z$ is the (left) principal eigenvector of $P_{\lambda, Z}$ and $p$ is the (unknown) initial state distribution of the Markov chain.

An upper bound on $\mathbb{P}(Z^T \hat{\mu}_{S_n} \in Z^T \Gamma)$ implies an upper bound on $\mathbb{P}(\hat{\mu}_{S_N} \in \Gamma)$ for $N \gg n$. For $\mathbb{P}(Z^T \hat{\mu}_{S_n} \in Z^T \Gamma)$ to meaningfully translate to $\mathbb{P}(\hat{\mu}_{S_N} \in \Gamma)$, we need to ensure that the sum of the three terms is positive.

The metric entropy term increases exponentially as $\Theta(\frac{d \log m}{n})$. As there is no other way to control this term,

we have to ensure that the dimension of the projection is much smaller than $n$, so that this term goes to zero. For controlling the sum of the remaining two terms, we have two parameters that we can choose: $\lambda$ and $Z$. The resulting optimization problem can be written as:

$$\sup_{\lambda \in \mathbb{R}^d, Z, d \ll n} (\langle \lambda, Zq \rangle - \log \rho_Z(\lambda)$$
$$- \quad \frac{1}{n} \cdot \max_{i \in [m]} (\log(\frac{p(i)}{\pi_Z(i)}), \log(\frac{\pi_Z(i)}{p(i)}))) \qquad (1)$$

The expression in Equation 1, involving the principal eigenvalue of matrix $P_{\lambda, Z}$, is not amenable to efficient global optimization over $\lambda$ and $Z$ since it is non-convex in $\lambda, Z$ taken together, but convex in each of the two sets of variables when considered separately. However, it can be shown that alternating maximization, over the $\lambda$ and $Z$ variables, will converge to a stationary point (Bertsekas, 1999). Thus one can obtain an upper estimate of the probability $\mathbb{P}(\hat{\mu}_{S_n} \in \Gamma)$ in time polynomial in $n, m$.

Finally, we define our modified Neyman-Pearson test.

Given an observed empirical distribution $\hat{\mu}_{S_n}$ on $\{1, ..., m\}$, and two Markov chain transition matrices $P_1$ and $P_2$, we can chose a $m \times d$ projection $Z$ to define the projected empirical log-likelihood:

$$\hat{S}_n := \sum_{i \in [d]} \langle Z^T \hat{\mu}_{S_n}, 1_i \rangle \cdot \log \frac{\langle Z^T \hat{\mu}_{S_n} P_2, 1_i \rangle}{\langle Z^T \hat{\mu}_{S_n} P_1, 1_i \rangle}. \qquad (2)$$

For large $n$, the usual Neyman-Pearson test would have consisted of comparing $\hat{S}_n$ with a chosen constant threshold, say $\gamma$, and accepting the alternative hypothesis ($H_1 : \hat{\mu}_{S_n} \sim P_2$) if $\hat{S}_n > \gamma$ and accepting the null hypothesis ($H_0 : \hat{\mu}_{S_n} \sim P_1$) otherwise.

In our small sample case, i.e., $n \ll \tau, m$, the test becomes: (1) if $\hat{S}_n - \gamma > \beta_Z$ accept the alternative hypothesis, and (2) if $\gamma - \hat{S}_n > \beta_Z$ accept the null hypothesis; for an error term $\beta_Z$ that depends upon $Z$.

Clearly, we want to chose $Z$ so that $|\beta_Z|$ is small, as we can not accept the null hypothesis, nor the alternative hypothesis, if $|\hat{S}_n - \gamma| < |\beta_Z|$. The error $\beta_Z$ arises because of using a small number of samples, and can be bounded in terms of $Z$ and the co-ordinates of principal eigenvectors of $P_1$ and $P_2$ as follows.

The *height* of the principal eigenvector $\pi$ for a transition matrix $P$ is the ratio between its maximum and minimum co-ordinates. Note that the contribution of the initial state term can be upper bounded by a quantity proportional to the log of the height of the principal eigenvector $\pi_Z$. In turn, the height can be upper bounded using the bounds (see for

---

[1] Note $1_j$ is a vector with 1 in the $j^{th}$ co-ordinate and 0 elsewhere.

example (Minc, 1970)):

$$h(P, \pi) \quad \leq \quad \frac{R_M - R_m}{\min_{i,j \in [m]}(P(i,j))}, \qquad (3)$$

where $R_M$ and $R_m$ are the largest and smallest column sums of $P$.

Under the assumption that $\min_{i,j \in [m]}(P_k(i,j))$ are comparable for $k \in \{1, 2\}$, and $\max_i p(i) \gg \min_i \pi_Z^{P_1}$, $\max_i p(i) \gg \min_i \pi_Z^{P_2}$, and $\max_i \pi_Z^{P_1} \gg \min_i p(i)$, $\max_i \pi_Z^{P_2} \gg \min_i p(i)$, we can use the previous discussion to show:

$$\beta_Z \leq \tfrac{1}{n} \cdot \max_{i \in [m]} \left( \log\left( \frac{\pi_Z^{P_2}(i)}{\pi_Z^{P_1}(i)} \right), \log\left( \frac{\pi_Z^{P_1}(i)}{\pi_Z^{P_2}(i)} \right) \right), \qquad (4)$$

where the RHS is $O(\log h(P_1, \pi_Z^{P_1}) + \log h(P_2, \pi_Z^{P_2}))$. Therefore, there's a trade-off in the choice of $Z$: whether to minimize the height or to optimize the projected empirical log-likelihood.

*Total error*: By definition, the total error in the Neyman-Pearson test corresponding to Equation 2 is simply: $\mathbb{P}_{P_1}(Z^T \hat{\mu}_{S_n} \in Z^T \Gamma) + \mathbb{P}_{P_2}(Z^T \hat{\mu}_{S_n} \in Z^T \Gamma)$, for some subset $\Gamma$ that separates $P_1$ and $P_2$ when the number of samples $n$ is large.

*Tying theory to application*: As a direct application of the above ideas we can design a test that uses the RNA expressions from a small number of cell samples from a tissue to test whether RNA expressions in the original tissue follow a prescribed transition matrix or not. More concretely, we sample the RNA expressions corresponding to the gene Cpe from a small number of beta cells ($n = 10$) from the murine pancreatic single cell data of (Bastidas-Ponce et al., 2019) [2] . We estimate the transition matrices corresponding to the Markov chain that models Cpe expression in the beta cells, alpha cells and ductal cells, see Figures 3 and 4. This was done by sorting the cells by their latent time (see (Bergen et al., 2020)) and then estimating the probability that a cell makes an immediate transition from Cpe expression $c_i$ to Cpe expression level $c_j$.

In this setting, we have the empirical distribution corresponding to Cpe $\hat{\mu}_{S_n}$ ($n \simeq 10$) sampled from the beta cells. We have the transition matrices $P_1$ (estimated from the beta cells), $P_2$ (estimated from the alpha cells), and $P_3$ estimated from the ductal cells. Note that both beta and alpha cells express non-trivial amounts of Cpe which is required for synthesizing and secreting hormones but not the ductal cells. Therefore, one would expect that with such a small number of samples, just 10 cells sequenced, it would not be possible to reliably distinguish between alpha and beta cells

---

[2]Their data is available at `https://scvelo.readthedocs.io/en/stable/scvelo.datasets.pancreas/` and `https://github.com/theislab/pancreatic-endocrinogenesis/`

using Cpe expressions but ti would be possible to distinguish between beta and ductal cells using Cpe expressions.

Indeed that turns out to be the case with our small sample Neyman-Pearson test (see Section 3 and Figures 1 and 2). The usual Neyman-Pearson test which assumes a large sample size, i.e., large $n$, can lead to fluctuating log likelihood values (see Figure 1). A more detailed set of experiments will be presented in a full version of this paper.

## 3. Further details of experiments on scRNA expression data

To verify the effectiveness of our algorithm, we design a test to check if we can correctly use just a few sample cells and expression levels for a single gene expression to distinguish between different cell types. We use the expressions for the gene Carboxypeptidase E (Cpe), required in the synthesis of hormones like insulin and glucagon, from scRNA sequencing data samples of murine pancreatic cells in (Bastidas-Ponce et al., 2019; Bergen et al., 2020).

Note that this is just a simplistic application illustrating how our test behaves with small sample sizes, and not an actual genetic test procedure, which would be far more complicated taking into account many biological and equipment related factors.

The set-up is described in this paragraph and our initial conclusions are described below. We use Cpe expressions from about 1000 pancreatic cells of three types: insulin secreting beta cells, glucagon secreting alpha cells and non-secretory ductal cells. We sample the expression levels of Cpe from random samples of $n \in [10, 20]$ beta cells, and construct our sample empirical distribution for each choice of $n$. We infer the transition matrices $P_1$ and $P_2$ for the Cpe secreting gene, from expression data for the secreting cells and non-secreting cells respectively, by sorting cells with latent time (Bergen et al., 2020) and computing the underlying Markov chain for Cpe expressions (see Figures 3 and 4), and then compute the log-likelihoods.

Our plots (Figures 1 and 2) show that in the harder problem, when trying to distinguish beta and alpha cells based on Cpe expressions from 10-20 beta cells, the usual Neyman-Pearson test can give very variable results, while our projected (small sample) test scores are close to 0 and suggest that one can not reliably distinguish between them using just a 10 cell sample. On the other hand, if we replace the alpha cells by non-hormone secreting pancreatic cells (ductal cells) then the problem becomes easier, and the hypothesis test scores become positive suggesting that we should be able to distinguish between beta cells and ductal cells using Cpe expressions from a sample of just 10-20 cells.

*Conclusion*: We propose a modified Neyman-Pearson test and our results show initial gains in reliability by suitably modifying the empirical log-likelihood.

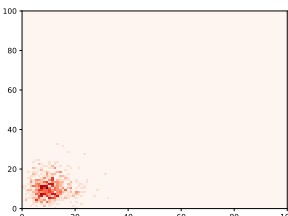

*Figure 3.* Heat map for the transition matrix of Cpe expression in beta cells. Darker areas indicate higher values.

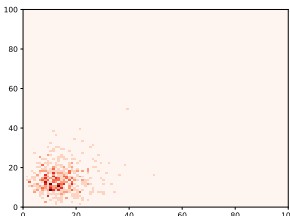

*Figure 4.* Heat map for the transition matrix of Cpe expression in alpha cells. Darker areas indicate higher values.

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
