# OpenReview forum: "Designing optimal tests for slow converging Markov chains"
_ICML.cc/2023/Workshop/IMLH — IMLH 2023 PosterShortPaper_

### Official Review · Reviewer_e9ur · 2023-06-15
**A modified Neyman-Pearson test for small sample number with application to distinguishing cell types**

**Rating:** 6
**Confidence:** 3

**Review:**

This paper presents a Neyman-Pearson test for differentiating between two Markov Chains with a small number of samples. It uses the proposed method to distinguish between different cell types with a few sample on scRNA expression data.

The topic is very interesting, and the results are promising. I vote for acceptance for this paper, but it would be better if the explanation of the empirical results can be more clear. For example, how to interpret Fig. 3 and Fig. 4?

---

### Official Review · Reviewer_G5gR · 2023-06-17
**optimal tests for markov chains**

**Rating:** 6
**Confidence:** 2

**Review:**

Positive points:
The paper introduces a novel Neyman-Pearson test for differentiating between Markov Chains using a small number of samples.
The authors propose a modified empirical log-likelihood approach and a projection-based technique to improve the reliability of the hypothesis test score.
The theoretical explanations are clear, and the application of the test to scRNA-seq data demonstrates its potential usefulness.
The paper acknowledges related work in Neyman-Pearson tests and provides insights into the limitations of assuming a large sample size for slow-mixing chains.

Negative points:
The proposed method requires further validation and experimentation to assess its generalizability and robustness.
The limitations of the projection-based approach and potential errors in the test due to information loss are discussed but not extensively explored.

Overall, the paper presents an innovative approach to Neyman-Pearson testing with limited samples and provides valuable theoretical insights. However, more empirical validation and thorough analysis of potential drawbacks would strengthen the findings.

---

### Meta-Review · Area_Chair_bRd3 · 2023-06-20

**Recommendation:** Accept (Poster)
**Confidence:** 5

**Metareview:**

Evaluating the machine learning system's effectiveness is critical for healthcare. In this paper, the authors propose a Neyman-Pearson test for inferring whether a given distribution of pancreatic RNA corresponds to a given transition matrix.  While the reviewers acknowledge the need for further validations to justify the proposed approach, they unanimously agree that the paper brings sufficient novelty and contribution to the community. With the concise yet impactful content of a short paper track, I am confident that the audience will appreciate the inclusion of this method.

---

### Decision · Program_Chairs · 2023-06-20

Accept (Poster Short Paper)